# Community perceptions and acceptance of ivermectin for malaria control on Sumba Island, Indonesia

Diana Timoria[1,2], Christa Dewi[3], Claus Bøgh[2], Tri Baskoro[3], Wisnu Nurcahyo[4], Vincentius Arca Testamenti ![ORCID][3], Lorenz von Seidlein[5,6], Kevin Kobylinski[5], Mary Chambers ![ORCID][7,6]*

1 Oxford University Clinical Research Unit, Jakarta, Indonesia, 2 The Sumba Foundation, Sumba Island, Indonesia, 3 Center for Tropical Medicine, Faculty of Medicine, Public Health and Nursing, Universitas Gadjah Mada, Yogyakarta, Indonesia, 4 Department of Parasitology, Faculty of Veterinary Medicine, Universitas Gadjah Mada, Yogyakarta, Indonesia, 5 Mahidol Oxford Tropical Medicine Research Unit, Mahidol University, Bangkok, Thailand, 6 Centre for Tropical Medicine and Global Health, Nuffield Department of Clinical Medicine, University of Oxford, Oxford, United Kingdom, 7 Oxford University Clinical Research Unit, Ho Chi Minh City, Vietnam

* mchambers@oucru.org

## Abstract

### Background

Indonesia has made significant progress in malaria control, however hotspots such as Sumba Island continue to experience high rates of malaria transmission, driven by multiple *Anopheles* mosquito species. The Sumba Livestock Ivermectin for Malaria (SLIM) trial was conducted to assess the efficacy of ivermectin treatment in livestock as a vector control strategy. This accompanying social science study aimed to explore community perceptions of ivermectin-based malaria interventions, including ivermectin livestock treatment (ITL) and potential mass drug administration (MDA) in humans.

### Methods

A social science study was conducted alongside the SLIM trial between November 2022 and September 2023 across four villages in Southwest Sumba. Qualitative and participatory approaches were used to explore community perceptions. 75 individuals (>18 years old) from the SLIM study village sites were included in four focus group discussions (59 individuals) and 16 individual in-depth interviews. We also held four feedback meetings for all members of each study village. Community engagement activities, such as puppet shows and interactive sessions on malaria transmission, were also implemented. Data were transcribed, coded, and thematically analysed using Nvivo software.

**Data availability statement:** Data is available through request to the Oxford University Clinical Research Unit data sharing committee and Card sorting full dataset is available at https://doi.org/10.5281/zenodo.16992379.

**Funding:** The Joint Global Health Trial (MR/V004670/1) UK funders including the Department of Health and Social Care, the Foreign, Commonwealth & Development Office, the Medical Research Council, and Wellcome. The funders had no role in the design of the study and collection, analysis, and interpretation of data or writing the manuscript.

**Competing interests:** The authors declared that no competing interests exist.

## Results

A total of 75 individuals participated in qualitative data collection, and approximately 650 individuals engaged in community events. Malaria was ranked as the most pressing health concern by study participants. Initial skepticism about ivermectin treatment of livestock was mitigated through trust-building efforts such as village meetings and respectful communication. Community members actively contributed to the trial, demonstrating acceptance of ivermectin treatment of livestock and expressing interest in future research participation. While there was openness to mass drug administration for humans, concerns about safety, particularly for children, were raised. Local authority approval was deemed essential for intervention acceptance. In one village, low social cohesion posed barriers to research participation, highlighting the importance of engagement before the trial started.

## Conclusions

Community perceptions of ivermectin-based malaria control strategies were shaped by trust, engagement, and cultural considerations. In this case there were multiple engagement activities built into the study – before, during and after the research, with stakeholders in local government and village leaders as well as with animal owners, other community members and children. The findings of this social research underscore the need for, and benefits of, sustained, respectful communication, partnership with local leaders and inclusive stakeholder engagement in malaria research. Future malaria control interventions should also account for local social dynamics, ensuring informed community participation to enhance trial feasibility and acceptance.

## Introduction

Since Indonesia established it's National Malaria Eradication Unit in 1952, the country has made great strides in reducing the burden of this parasitic, mosquito-borne disease [1]. Despite a spike in cases since 2023, up to 89% of the national population now live in malaria-free areas [2]. The areas of high endemicity are concentrated in West Papua, the province of Nusa Tenggara Timur and difficult to reach populations in Sumatra, Kalimantan and Sulawesi [3].

Leveraging such progress, the government aims to eliminate malaria nationwide by 2030. This appears an ambitious goal, especially for resource limited areas such as eastern Indonesia, which have disproportionately high burdens of malaria. Malaria elimination has proved particularly challenging on Sumba, an island belonging to the eastern province of Nusa Tenggara Timur. In 2022, the island recorded an annual parasite incidence API of 16.6, which was more than 10 times the national average. In 2022, a total of 13,262 clinical malaria cases were documented on Sumba [2]. At least 12 species of *Anopheles* mosquitoes are present on Sumba [4,5], each with differing feeding behaviours, making vector control particularly difficult. No *Anopheles* host preference studies have been conducted on Sumba prior to this study [6].

However, given the significance of large animals to Sumbanese residents [7–9], with more than 65% of households owning livestock [10], the island presents an opportunity for testing novel vector control measures by using livestock endectocides. Research from Burkina Faso, South Africa, Kenya, Pakistan and Vietnam has shown that after cattle are injected with ivermectin, their blood is lethal to blood-feeding *Anopheles* [11–15]. Ivermectin treatment of livestock (ITL), if applied *en masse*, may substantially disrupt malaria transmission on Sumba.

With ITL as a tool for reducing malaria transmission, a veterinary trial entitled *Sumba Livestock Ivermectin for Malaria control (SLIM)* was implemented on the island from 2022–2023. The trial's primary objective was to treat cattle and buffalo with short and long-lasting ivermectin formulations and evaluate the impact on the survival of wild *Anopheles* [6].

Community involvement and acceptance of malaria control interventions is essential, particularly in the case of mass drug administration (MDA) efforts [16]. Coverage in target populations should exceed 80% to interrupt local malaria transmission [17–19]. In addition, MDA may provide a significant selective pressure for emergence of resistance and therefore should not be applied unless there is a high chance of community acceptance and that elimination is feasible [16]. To date there are no studies that report community perceptions of the use of ivermectin for malaria control through treatment of livestock or MDA in humans. Alongside the SLIM entomology trial, we conducted a social-science study to understand community acceptance of ivermectin for malaria control. More specifically, we aimed to explore perceptions surrounding the use of mass ivermectin treatment for livestock (ITL) and ivermectin mass drug administration (MDA) to humans.

## Methods

### Study setting and site selection

Our study took place on Sumba, part of East Nusa Tenggara province, eastern Indonesia (see Fig 1). It is an island of around 11,000km² [20], which was inhabited by close to 800,000 people in 2022 [21]. In the same year, it remained one of poorest islands in Indonesia, with an overall poverty rate of 29% [22]. Most Sumbanese adults have only attained a primary-school education and engage in agriculture for a living [23].

As a subset of agriculture, livestock farming is a means for many island residents to access food, cash, and even social prestige [7–9]. Of the 150,000 households here, over 90,000 families raise large animals, including cows and buffaloes.

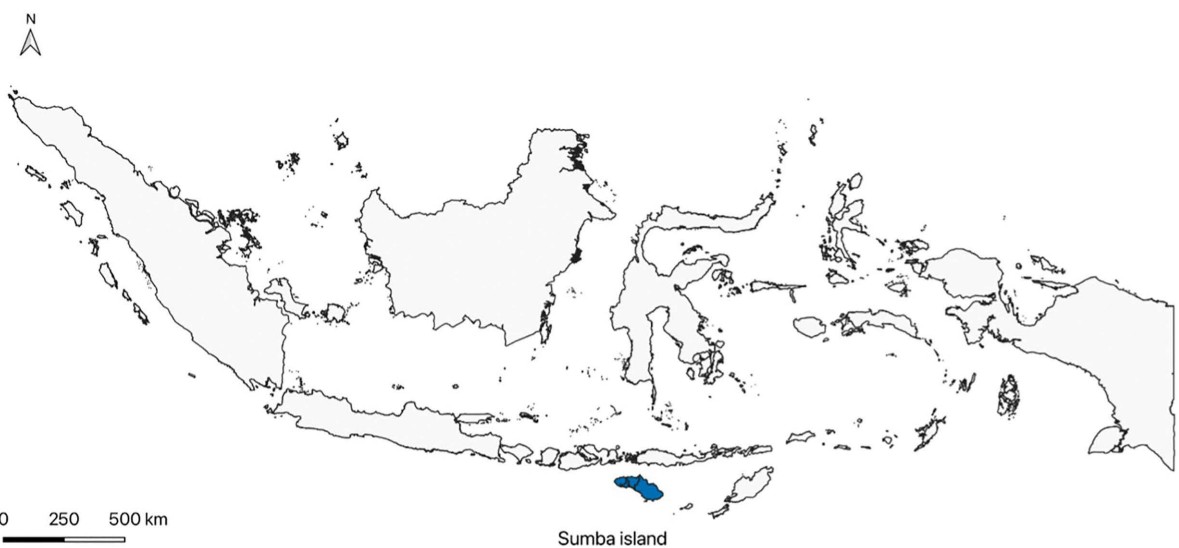

**Fig 1. Map of Sumba Island in eastern Indonesia.** Created in QGIS [24] using administrative boundary data from GADM version 4.1 [25].

Most keep livestock under their houses [26], which poses as a risk factor for malaria in humans [27]. Much of the malaria transmission on the island occurs in the West and Southwest districts of Sumba [28]. These two districts are largely dominated by the Kodi ethnic group - the largest group in the western part of the island , least educated and resourced, and has the highest malaria burden.

Within this socio-economic context of Sumba, the SLIM trial and the current social-science study approached six villages in the Southwest (Sumba Barat Daya) and West (Sumba Barat) districts of Sumba. These villages were selected due to their *Anopheles* species diversity and abundance. Two villages turned out to be unfit for investigation. The first village had a damaged mosquito larval habitat following heavy rains from a passing cyclone and a flash flood one month later and thus did not have enough mosquitoes in circulation to be a valid site. The second village presented what we considered an unwillingness to participate in research (more details in the **Results** section).

We were thus left with four villages to conduct fieldwork. Our final site selection included Pandawawi (subvillage Kahale), Matakapore, Waimakaha, and Wainyapu which included two sub-villages (Galukoloko and Waikavaroko), which were known to have a high diversity of *Anopheles* species. Administratively these villages fall under Southwest Sumba District, each having an average population of 1,500 and an area of approximately 7km$^2$ [29–33].

## Study design and data collection

From 16th November 2022–31st December 2023, entomology fieldwork was conducted in four villages, spending approximately 3 months/site. The methods for the SLIM trial are published elsewhere [6]. The current social science study accompanied the SLIM trial with community engagement and systematic data collection in every village, following the same timeline.

Pre-study sensitization was conducted at district and sub-district level, principally with key stakeholders such as the local government and village leaders, local health services and local veterinarians. The community engagement components included whole village meetings plus activities such as puppet shows, board games, drawing/coloring, and viewing mosquitoes under the microscope. These took place once the study site was agreed and after the initial focus group discussions, card sorting activities and interviews had taken place, so as not to influence the data about knowledge and perceptions of malaria and research. Once this data had been collected engagement activities continued throughout the study period (3 months per village). In addition, during the trial mosquitoes that had fed on ivermectin-treated and untreated livestock were presented to the community so they could personally observe the poisoning effect ivermectin had on the dying mosquitoes. These activities aimed to create opportunities for open discussion with the study communities, to sensitize community members about the SLIM trial and to promote better understanding of malaria (it's symptoms as well as transmission). We organized community engagement twice in each village, before and after the entomology collections, running a total of eight community-engagement events.

Recruitment for the qualitative research component was purposive to include adults (>18yo) from the study villages. Village leaders identified 8–10 men who were animal owners, plus 2–4 older women, with experience handling livestock, who they felt would be willing to speak in a mixed gender focus group discussion (FGD). They also introduced 4 women who had experience handling livestock to take part in the in-depth interviews IDIs (including women who may have been less confident to speak in a mixed FGD). These people were invited to a study meeting where the requirements of the study were explained. Those agreeing to join the study gave written informed consent. The data collection component included three primary activities. First, we conducted focus group discussions (FGDs) with various village stakeholders to explore their experiences and views of malaria, as well as their possible acceptance of malaria control by ITL (as seen in the SLIM trial), and hypothetically for human MDA to control malaria. During the FGDs we also asked participants to undertake card-sorting, i.e., ranking health issues according to their levels of importance, where each card represented a common disease or ailment on Sumba. Since the participant group included village leaders and local government officials, the FGD was facilitated carefully, aware of social hierarchies, to ensure all participants were encouraged to speak.

Second, we carried out in-depth interviews (IDIs) with women (>18 years old), because they were generally less comfortable speaking in the FGD settings, using a semi-structured interview guide. The interviews took place in a location convenient for the participant, usually on the veranda of their house. Finally, we organized feedback meetings (FMs) after the completion of the SLIM entomological collections, the purpose of which was to seek their additional insights following the conclusion of the trial activities. In each village, we conducted one FGD, four IDIs, and one FM. This amounted to a total of four FGDs, 16 IDIs, and four FMs in all study sites. The three primary activities of data collection were supplemented with participant observations, which was documented as field notes when the authors joined various activities under the SLIM trial and its associated social-science study.

Data collection was conducted in Bahasa Indonesian by the author DT, with support of a local Kodi language speaker who translated if participants did not speak Bahasa. The FGDs and IDIs were recorded and transcribed in Bahasa. The community engagement activities with children were conducted by two local volunteers (teachers) in Kodi language. The engagement activities with adults were conducted in both Indonesian and Kodi.

### Data analysis

We transcribed all focus group discussions (FGDs), in-depth interviews (IDIs) and feedback meetings, making them available as 24 transcripts. The transcripts were uploaded onto NVivo (a qualitative data analysis software package) (Lumivero, version 14) [34], for thematic analysis, following the specific guidelines of Braun and Clarke (2006) [35]. Initial coding was conducted by authors DT and CD, and triangulated with a third person (MC). After the three researchers agreed on a coding frame, DT coded all transcripts in Bahasa Indonesian and translated a set of illustrative quotes into English. Codes were organized into themes based on the guides prepared for FGDs, IDIs and feedback meetings, with sub-themes added as they emerged.

Card-sorting data from the FGDs was captured by photographing the sorted lists, transferred to a spreadsheet, and analysed using Microsoft Excel.

Participant observations, in the form of field notes, were discussed among authors, as well as with the wider trial teams. They were used to verify and extend the analysis already conducted from FGD, IDI and feedback meeting data.

### Ethics declaration

SLIM's human-study protocol covered the current social-science study and received approval from the ethics committees of the Medical and Health Research Ethics Committee, Faculty of Medicine, Public Health and Nursing, University of Gadjah Mada (KE/FK/0773/EC), Indonesian National Research and Innovation Agency (BRIN) (023/KE.02/SK/8/2022), and the Oxford University Tropical Research Ethics Committee (556‒21).

Local permission was granted by the government of Southwest Sumba District; the governments of Kodi Balaghar and Kodi Bangedo Sub-Districts; and community leaders of Pandawawi, Matakapore, Waimakaha, and Wainyapu Villages.

All FGD, IDI and FM participants provided written consent.

## Results

This section summarizes our analyses of FGD, IDI, FM and participation-observation data. Results are organized into six domains: (1) Participants characteristics, (2) Common health problems, (3) Villagers experiences and views of malaria, (4) Community perceptions of ITL, (5) Community perceptions of ivermectin MDA, and (6) Possible barriers to malaria research.

### Participants characteristics

Our community engagement events attracted an estimated 650 participants. Over 40% of them were primary school children (7–13 years old), while the rest were adults (> 18 years old).

Our data-collection activities enrolled 75 participants, including 59 who joined FGDs and FMs (the same individuals were invited first for FGDs and again for FMs), and 16 (female) who answered IDIs. People contributing to the current social-science study were a mix of those who directly participated in the SLIM trial (having their livestock receive ivermectin or serve as a control), and those who only knew about the trial. Participation in the social-science study was spread evenly among Pandawawi, Matakapore, Waimakaha and Wainyapu Villages, each contributing 14–22 participants. In terms of village roles, the stakeholders included government officials, community leaders, livestock owners, and health cadres, the latter of whom constituted the largest group (23 participants). Because participants were selected based on their experience with livestock (owning cows or buffalo or horses), the participants were all over 25yo and mainly male. 32 participants (less than half) were female, reflecting gender differences in roles regarding livestock. Table 1 provides details of the participants' characteristics.

## Common health problems

All 59 FGD participants took part in the card sorting exercise. However, most of them worked in pairs (54) and only 5 participated as individuals, thus producing a total of 32 responses.

Each response was a set of 12 cards representing 12 common health problems on Sumba (as suggested by local healthcare professionals). When a health problem was identified by a pair/individual as the most important, it received a score of 12. Less important problems received scores lower than 12. The least important health problem was coded as 1.

We organized all 32 responses in a spreadsheet (see Supplementary Material) and aggregated the score for every health problem investigated at the FGDs. Table 2 is the full list of 12 common health problems on Sumba and their associated ranking by the study participants. The full dataset is available as supporting information S1.

Collectively FGD participants identified malaria as the number one disease troubling Kodi residents. Malaria's score was 332, which was 42% higher than the score given to influenza, the second-most important health problem on the list. Malaria was considered three times as important as Covid-19, which FGD participants deemed to be the least concerning in 2022–2023.

**Table 1. Characteristics of participants (N = 75). Data collected on Sumba Island between November 2022 and September 2023.**

| Characteristics | Focus Group Discussions and Feedback Meetings | In-Depth Interviews | Total (%) |
|---|---|---|---|
| *Village* | | | |
| Pandawawi | 10 | 4 | 14 (18.7%) |
| Matakapore | 17 | 4 | 21 (28.0%) |
| Waimakaha | 18 | 4 | 22 (29.3%) |
| Wainyapu | 14 | 4 | 18 (24.0%) |
| *Role* | | | |
| Government officials | 14 | 0 | 14 (18.7%) |
| Community leaders | 16 | 1 | 17 (22.7%) |
| Animal owners | 16 | 5 | 21 (28.0%) |
| Health cadres | 13 | 10 | 23 (30.7%) |
| *Gender* | | | |
| Female | 16 | 16 | 32 (43.0%) |
| Male | 43 | 0 | 43 (57.0%) |

**Table 2. Common health problems on Sumba ranked by study participants.**

| Health problem | Score |
|---|---|
| Malaria | 332 |
| Influenza | 234 |
| Skin diseases | 232 |
| Snake bites | 227 |
| Worms | 226 |
| Diarrhea | 214 |
| Dengue | 209 |
| Tuberculosis | 193 |
| Measles | 168 |
| Injuries | 150 |
| Tetanus | 136 |
| Covid-19 | 107 |

### Villagers' experiences and views of malaria

All 75 individuals who formally contributed data at FGDs, IDIs and FMs were familiar with malaria. Either they had contracted malaria before, or knew someone afflicted by this disease. Having multiple malaria cases in the same family was not rare. A woman mentioned that almost all of her children had had malaria, including her twins and youngest child. The vast majority of participants were able to name malaria symptoms, including most commonly fever, dizziness, and body ache.

Many villagers were aware that mosquitoes transmitted malaria. Some were quite specific when they named *Anopheles* as the vector, pointing out that their population tended to surge around the end of each year, or the beginning of Sumba's rainy season. *"Also, when it is already November and December, when the rainy season begins, there start to be many Anopheles mosquitoes. Sometimes they are found in swamps or puddles where there are many larvae. Those are usually malaria mosquitoes."* (Participant from FGD02V04).

As for treatment, we did not observe any dominant pattern among the participants. Villagers' stories seemed equally split between early treatment and delay in healthcare seeking. Those who were quick to seek medical attention reported going to community health centers (government-run) or local clinics (privately-operated). *"As for me, whenever someone from my house starts to feel a fever or shivering and vomiting, I take them to the clinic on the same day. And if the queue is too long, I take them to the community health center. There, if they are examined and diagnosed with malaria, they usually get medicine, and if they take it until it's finished, they feel refreshed again."* (Participant from FGD01V03). Those who delayed seeking formal medical care reported trying to control disease symptoms themselves, by taking medications (e.g., paracetamol) or home remedies (e.g., mahogany seeds and papaya leaves). One participant brought up the case of a mother who took her son to the community center when he was already having convulsions. The boy tested positive for malaria and showed complications for several diseases. His case was so severe that the center's staff referred him to a hospital in Southwest Sumba District.

Overall, the participating villagers appeared confident with their own knowledge of malaria, as well as good community awareness of the disease. They attributed such awareness to the health-promotion efforts of a wide range of entities, such as community health centers, Perdhaki (the Association of Voluntary Health Services of Indonesia) and local health NGOs. *"So we [Perdhaki] also help the community health centers in providing education about malaria through 'village discussions'. We explain how we can deal with mosquitoes for example, by planting citronella, and also what causes malaria, how malaria is transmitted, and what the signs and symptoms of malaria look like."* (Participant from FGD02V02).

## Community perceptions of ivermectin and ivermectin treated livestock (ITL)

Perceptions of ivermectin and ITL were sought from 75 participants of FGDs, IDIs and feedback meetings. The FGDs were conducted before any sensitization had been given, whilst the IDIs and feedback meetings were held towards the end of the trial or afterwards. The comments made applied specifically to SLIM, rather than a hypothetical intervention treating animals with ivermectin to control human malaria.

Some animal owners recalled feeling skeptical at first, for they feared that injecting cows and buffaloes when they were not sick would cause harm to the livestock . However, they decided to have their livestock enrolled in SLIM partly because the research staff explained all trial procedures to be safe, and mostly because there was a government veterinarian from Southwest Sumba present on the team. *"As for us as animal owners, we put our trust in the research team because the team includes a veterinarian who can take responsibility for the animals' safety and health. The research team also always comes, and the entomology team always communicates with us every time they want to conduct mosquito collection. So, we feel safe and are happy to be able to support this research."* (Participant from FGD02V04).

Many community members, even those who did not have animals enrolled, came to appreciate SLIM's working approach. Without making a distinction between the SLIM trial and it's associated social-science study, they described communication from the researchers as simple, regular, and respectful. A woman compared SLIM with a previous malaria study she had experienced, and observed the SLIM trial to be more positive for her community: *"I think that when this team did the research, you have taken a friendlier approach all this time. Very respectful of culture in terms of your work and attitude, conveying information to us for the whole study time. The community understands this research, and I am sure that the community supports this. You know how to provide information to them so that they can easily understand, especially the information before the study"* (Participant from IDI01V02).

Showing their acceptance of the trial, community members provided both material support and specific advice to help the research team operate trial activities. For example, some villagers contributed bamboo to install net traps for collecting mosquitoes that landed on SLIM livestock. In Pandawawi village, many locals were involved in the setting up of a net trap, advising researchers to place it further away from the sacred tree of their community. A significant part of Kodi's population practice Marapu, a belief system indigenous to Sumba, designating certain trees and megalithic structures as sacred. Placing research instruments such as SLIM's net traps close to these sacred entities would have been considered culturally inappropriate.

Community members also expressed satisfaction from the intended trial outcome, i.e., mosquitoes dying after they blood fed on ivermectin-treated animals. *"Yesterday, during the activity [engagement] at the church, the research team brought samples of mosquitoes that had died in this study. I was happy to know that it worked. So it wasn't in vain for the community here to lend their animals."* (Participant from IDI02V02).

Several IDI participants communicated a sense of pride, seeing that they could contribute to science and help with malaria control in their communities. One woman went as far as to register her willingness to participate in future ITL projects. *"I support this research. I also have a younger sibling whose animals are used in this research, including a buffalo and a cow. I am proud of him because he wants to help. If I have an animal that can be used, I will give it for the research. Hopefully, there will be more research in the future, and if I can get involved, I will help too. Which is important for our health in the village"* (Participant from IDI03V04).

## Community perceptions of ivermectin mass drug administration (MDA)

To understand perceptions surrounding the use of ivermectin MDA in humans, we also asked the participants to consider whether they would accept the drug for themselves, or their children if it became available in the future. The difference was that they could only comment on MDA in a hypothetical sense, whereas what they had communicated about ivermectin used in livestock was tied to their experience of the SLIM trial happening in their own villages.

Although we asked generally about possible MDA using ivermectin to control malaria, several FGD participants assumed that the same SLIM trial team would expand their activities to mass drug administration with ivermectin for Sumba residents. The interviewers were careful to explain that there was currently no permission for this yet. However, the participants were willing to support MDA interventions, given their positive experience with the SLIM trial. *"If there are more, we already know you and the team, we have already participated, so I think that is fine, we can still support you"* (Participant from FGD02V01).

The trust implied above was far from being unconditional. Instead, most participants of the FGDs, IDIs and FMs exercised caution when they said MDA (and any new research projects) would need to seek permission from local health authorities. Ivermectin currently has limited approval as human drug in Indonesia, and thus people on Sumba are unlikely to have had past experience with this medication. There was little Indonesian media discussion of ivermectin as a Covid-19 drug during the pandemic.

Finally, some FGD participants held a special reservation over who should serve as a target for MDA. Three people in Wainyapu Village emphasized that they would accept ivermectin for themselves, but not children. *"Let me try it. Do not go to the children. I am afraid the drugs might be toxic for them"* (Participant from FGD01V04).

## Observed barriers to malaria research

Our social-science study, alongside the SLIM trial, approached six villages but ended up conducting fieldwork in four. We withdrew from two villages: The first one had its mosquito habitat damaged following heavy rains. The second featured an environment suitable for SLIM trial procedures, but researchers from both the trial and social-science sides felt that collectively speaking, the community was unwilling to participate.

Through observation and stakeholder feedback we identified that the community in the second village posed some barriers to not only SLIM, but also research more generally. The first issue raised was regarding the level of compensation offered. At first, several livestock owners in this village agreed to the standard rate with which SLIM would compensate it's trial participants. Later, they said the compensation would need to be higher and refused their animals. People who had promised to look after the livestock enrolled in the study also suddenly refused to do so.

In screening livestock from this site for potential inclusion in the SLIM trial it was discovered that several buffalo had Surra (Trypanosomiasis) which is a particularly devastating disease for horses, which have great cultural significance on Sumba. As this village was outside of the Kodi subdistrict, the study team collaborated with government health veterinarians that spoke the local Gaura language, and offered free diagnosis and treatment for this disease, which was still refused by some livestock owners. Furthermore, there were rumours about a buffalo that had died. Although the SLIM trial had not injected any livestock in that village, and the cause of death of the animal was unknown to the study team, this death may have frightened animal owners away from participating.

We aimed to investigate the root cause of this increasingly unwelcome atmosphere, but could not find any villagers in Gaura subdistrict willing to explain the real situation. However, we later gathered (from a study team member who was local to that area), that there had been a conflict between different neighborhoods within the village, specifically the residents closest to the mosquito habitat that SLIM trial wanted to work in. Such lack of social cohesion seemed to be a contributing factor as to why the research permission given by community leaders did not represent strong collective consent from the village members. We were advised by local staff that the community leaders had not consulted with the wider community.

## Discussion

In congruence with official statistics [2], the community feedback confirmed that malaria remained a top concern in the Kodi area of Sumba, and its importance surpassed other health problems by a wide margin.

To address this important health problem, participants of the current social-science study expressed a nuanced acceptance of ivermectin for malaria control. Community members maintained that interventions in the forms of ITL for animals and MDA for humans should be used for livestock and adults (but were hesitant about giving it to children), should have right approval from a respected health authority, and the right expertise for functioning in the field (e.g., veterinarians and doctors). These insights were valuable and yet unsurprising, given that they constitute the basic elements of research protocols anywhere in the world. What seemed novel from the research on Sumba was that many participants articulated the need for simple, regular, and respectful communication, and in the case of one particular village, we observed a lack of social cohesion as possibly hindering trial activities.

We believe that community members appreciated the working approach taken by SLIM's trial and social-science researchers, which in turn triggered active contribution on the part of Kodi residents. This reality contrasts the current literature, which tends to portray the islanders in a negative light. In government accounts, ordinary Sumbanese are oftentimes seen as passive recipients of assistance (e.g., mosquito nets) [36]. Guntur et al (2021 and 2022) report poor malaria knowledge on Sumba and attribute it to low levels of education [37,38]. Mariana and Martha (2024) also follow the knowledge deficit model, highlighting Sumbanese women's superstitious beliefs about malaria [39]. A 2024 report commissioned by UNICEF depicts an attitude of complacency among island residents, who find it hard to grasp scientific ideas such as asymptomatic malaria [40]. Our study demonstrated that the communities had sufficient knowledge about malaria and the vectors that when we took time to explain about the concept of MDA, using pictures and stories, they were able to engage in active discussions about the possibility of using mass drug administration of ivermectin for malaria control. Other studies in the Greater Mekong Subregion also report that rural communities can engage in discussing complex ideas such as MDA if the effort is made by researchers to make these concepts understandable [19,41]. Furthermore, communities within a region may have different levels of understanding and interest, therefore engagement should be preceded by social science research so that strategies can be tailored to local needs [41].

What we came to realize in 2022–2023 was that Kodi residents knew sufficiently about malaria so that when SLIM researchers communicated with simplicity, regularity and respect, community members engaged in discussions and reciprocated by making an active contribution to research. While this reciprocity may seem instant, it is likely that trust had been built over a long time because two of the SLIM partners had researched malaria on Sumba since as early as 2001 (Sumba Foundation) [42] and 2010 (Oxford University Clinical Research Unit) [43–48]. The Sumba Foundation partners were well known and trusted for their healthcare clinics and social projects. An interviewee implied that her community previously did not want to contribute when there had been a different malaria trial that was unfriendly and treated research as a one-off event. Researchers working on MDA studies in Laos and Cambodia found similar mistrust based on the communities past experiences with NGOs that 'had not kept their promises' [17]. The good communication and trust that the SLIM team had demonstrated, i.e., simple, regular and respectful, should therefore be taken up in malaria research as a tool for motivating active contribution from community members, regardless of the economic and educational settings they find themselves in. Pell et al 2019, report that trust in the context of a mass drug administration research study in the Greater Mekong sub-Region was strengthened by the involvement of local health staff and other authority figures (teachers, village leadership etc.) in the study activities. In a similar way in the SLIM trial communities, the animal owners accepted and trusted the government veterinarians because of their expertise, and because they previously supported cattle owners. In this context, government officials were trusted and were an asset to the study, however in other settings where communities mistrust the government, their involvement might hinder study-community relationships [17]. Study teams should assess these social dynamics before inviting them to join study teams.

On lack of social cohesion, there is a history of animosity on Sumba, which is not along religious or ethnic lines, but which tends to be clan-based [49]. Legal scholar Jacqueline Vel argues that in certain parts of the island, there is hardly a single leader that unifies all community members. Different factions in the government may rely on their respective clans

to fight each other, and vice versa [49]. The implication of this tradition is that when a trial has obtained approvals from all levels of government, and yet there is an internal conflict between different government leaders or different clans, one side may block research cooperation if they perceive the research to be beneficial for the other side only. Such a situation is not rare for studies in low- and middle-income countries [50]. Pell et al (2019), report that local community dynamics influenced community acceptance of MDA. In Myanmar, where the community perceived that the study was politically aligned, those who supported the opposing groups refused to participate [17]. Vincent et al's (2022) review of community engagement with malaria research recommends that where social cohesion is lacking, scientists should seek out a diversity of stakeholders and engage them using a different method appropriate for each group [50]. We agree with the solution proposed by Vincent et al (2022), and would add that in a fragmented community, research staff may do well by preceding trial activities with inclusive engagement over a long period of time. For SLIM, our plan was to do social science alongside the primary trial. If we had scheduled the collection of qualitative data and organization of community events as an extended lead time before trial activities, we might have uncovered ways to work with different community segments and conduct the SLIM trial in the village with low social cohesion.

The current paper has its strength and limitations. Our study was best at combining traditional methods (FGDs, IDIs, feedback meetings, and participant observations) and active listening at the population level through a series of community-engagement events as well as establishing relationships with local leaders and including local people into the study team. We were thus able to capture the nuances surrounding community acceptance of ivermectin for malaria control. On the other hand, our study had some limitations. First, some FGDs were too large. While standard research guidelines recommend a maximum of 12 participants/discussion [51], we conducted three FGDs that included 14–18 participants. Our plan was to have 10 participants/FGD, but some villagers came uninvited and we thought it would be against the spirit of community engagement to send them away. We still managed to learn meaningful insights from the large FGDs by having two facilitators for every discussion, where they took turns to ensure that all participants could express themselves freely without feeling left-out. Second, we had a total of 75 people contributing data, a sample probably too small to represent the entire village populations. However, complementary to this modest sample, our community-engagement events reached about 650 people, which was approximately 11% the combined population of Kahale, Matakapore, Waimakaha, and Wainyapu where the SLIM trial was implemented. If there had been views about ITL and ivermectin MDA that were contrary to the formal qualitative data, we might have recognized them at these community-engagement events.

As Indonesia is determined to eliminate malaria by 2030 and there remain several parts of the country that experience a high prevalence, there needs to be new tools to control this disease.

Preliminary studies on MDA in highly endemic areas of Papua Province [52], and the SLIM trial on the use of ITL [6], suggest promising potential as new malaria control strategies in Indonesia. We recommend that continued research into these approaches take a systematic approach to understanding the community contexts, involving local leaders and stakeholders, and communicating simply, regularly, and respectfully with community members for maximum acceptance.

We further add to the malaria literature that in areas of low social cohesion, researchers should engage a diversity of community segments, be ready to apply different approaches with different groups and sustain engagement over an extended period before implementing trial activities.

## Supporting information

**S1 Table. SLIM Social Science Supplementary Material.** Card sorting full dataset https://doi.org/10.5281/zenodo.16992379.
(XLSX)

**S2 File. Topic guide for focus group discussions.** Qualitative research tools. https://doi.org/10.5281/zenodo.17769585.
(DOCX)

**S3 File. Topic guide for semi-structured interviews.** Qualitative research tools. https://doi.org/10.5281/zenodo.17769585.
(DOCX)

## Acknowledgments

We acknowledge the partnership of SLIM participants, the wider communities and community leaders in Kahale, Mataka-pore, Waimakaha and Wainyapu Villages in Southwest Sumba. We acknowledge the support of local government in Kodi District and Southwest Sumba. We acknowledge the support of influential Kodi individuals who facilitated the study: Ferdi Mori, Karola Ina Kue and Felisia Delsi Bebon.

## Author contributions

**Conceptualization:** Mary Chambers, Christa Dewi, Claus Bøgh, Tri Baskoro, Wisnu Nurcahyo, Lorenz von Seidlein, Kevin Kobylinski.

**Data curation:** Diana Timoria.

**Formal analysis:** Mary Chambers, Christa Dewi.

**Funding acquisition:** Mary Chambers, Tri Baskoro, Wisnu Nurcahyo, Vincentius Arca Testamenti, Lorenz von Seidlein, Kevin Kobylinski.

**Investigation:** Diana Timoria, Kevin Kobylinski.

**Methodology:** Mary Chambers, Diana Timoria, Christa Dewi, Claus Bøgh, Tri Baskoro, Wisnu Nurcahyo, Lorenz von Seidlein, Kevin Kobylinski.

**Project administration:** Mary Chambers, Diana Timoria, Claus Bøgh, Vincentius Arca Testamenti, Kevin Kobylinski.

**Resources:** Vincentius Arca Testamenti.

**Supervision:** Mary Chambers, Christa Dewi, Claus Bøgh, Kevin Kobylinski.

**Writing – original draft:** Mary Chambers, Diana Timoria, Christa Dewi.

**Writing – review & editing:** Mary Chambers, Kevin Kobylinski.

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
