## [Decision Letter · Decision Letter 0]

3 Aug 2025

Dear Dr. Chambers,

Thank you for submitting your manuscript to PLOS ONE. After careful consideration, we feel that it has merit but does not fully meet PLOS ONE’s publication criteria as it currently stands. Therefore, we invite you to submit a revised version of the manuscript that addresses the points raised during the review process.

We look forward to receiving your revised manuscript.

Kind regards,

Pyae Linn Aung

Academic Editor

PLOS ONE

Journal Requirements:

3. We note that Figure 1 in your submission contain map images which may be copyrighted. All PLOS content is published under the Creative Commons Attribution License (CC BY 4.0), which means that the manuscript, images, and Supporting Information files will be freely available online, and any third party is permitted to access, download, copy, distribute, and use these materials in any way, even commercially, with proper attribution. For these reasons, we cannot publish previously copyrighted maps or satellite images created using proprietary data, such as Google software (Google Maps, Street View, and Earth). For more information, see our copyright guidelines: http://journals.plos.org/plosone/s/licenses-and-copyright.

4. Please remove all personal information, ensure that the data shared are in accordance with participant consent, and re-upload a fully anonymized data set.

Additional guidance on preparing raw data for publication can be found in our Data Policy (https://journals.plos.org/plosone/s/data-availability#loc-human-research-participant-data-and-other-sensitive-data) and in the following article: http://www.bmj.com/content/340/bmj.c181.long .

Additional Editor Comments:

Two reviewers have provided many useful comments and suggestions. Please carefully revise the manuscript accordingly, preparing a point-by-point response and a tracked-changes version to document all modifications made.

Reviewers' comments:

Reviewer's Responses to Questions

**Comments to the Author**

1. Is the manuscript technically sound, and do the data support the conclusions?

Reviewer #1: Yes

Reviewer #2: Yes

2. Has the statistical analysis been performed appropriately and rigorously?

Reviewer #1: N/A

Reviewer #2: N/A

3. Have the authors made all data underlying the findings in their manuscript fully available?

Reviewer #1: Yes

Reviewer #2: Yes

4. Is the manuscript presented in an intelligible fashion and written in standard English?

Reviewer #1: Yes

Reviewer #2: Yes

Reviewer #1: This paper by Chambers et al. describes the results of qualitative research conducted along an ivermectin treatment of livestock (ITL) in Sumba, Indonesia. Overall, additional information on certain aspects of the methods, and a more detailed conclusion section on how the results can be used to improve research in this context, would benefit the manuscript.

Please see below specific comments:

Introduction

The introduction is very well written and very informative.

Line 88: minor but suggest grouping this information (the specific number of cases) with the information in line 60.

Study design

It would be helpful to include very clear information on the timing of this study related to the SLIM project. Although implied that you did not work before the implementation of SLIM, it is only in the discussion section where this information is specified.

Minor, but suggest adding “author” to all instances where the authors initials are used

Results

Line 170: school children were all participants up to 18? 21? And then anyone over that age was considered adult? Seems that only having two categories here “school children” and “adults” would leave some participants out? I suggest adding information to clarify

It would be interesting to know the age distribution of your participants. Were there on the older side? Age can be a key determinant for perception on ‘new’ methods to control mosquitoes, and it would be helpful to know if all your participants were younger vs older adults.

Table 1: Add the total N to the table. Make sure that the title of the table is complete, including dates, and place, so that stands alone.

Line 198: Consider moving this to the discussion section

Line 227: More information on informational/educational campaigns that took place before the implementation of SLIM would be helpful to give more context to this part of the study. Was there a broad community outreach effort before this study?

Line 233: this seems to be a very important finding, that animal owners trust government veterinarians and their involvement in projects like these could improve participation, but I don’t think this is highlighted in the discussion. Government involvement is not always a factor that increases participation in research studies, and this would be an important context-specific finding.

Line 235: It is not clear to me what do you mean by “further into the trial”. Do you mean that in activities conducted at a later time, you got these findings?

Line 263: This study was conducted after the COVID-19 pandemic. Was ivermectin use also widely discussed in media and social media in Indonesia? If yes, did you get any insight on participants perceptions associated with the use of ivermectin for covid and how that translate to MDA?

Line 280: Just curious, I understand that you are translating directly from participants’ quotes, but was “drunk on drugs” the most accurate translation? Did they mean that drugs could be toxic for them? Or is being “drunk on drugs” a common saying in your population?

Line 283: This section doesn’t seem to be something that you assessed through your study activities, but rather a description of some situations observed during the activities conducted to recruit participants for the SLIM study. When I read the section where you describe the topics, I thought you were going to describe results of specific questions asked to participants regarding what they perceive as barriers for implementing malaria research. I suggest modifying your results section to clarify that this is additional information but not part of your study questions/assessment or to modify language to clarify how you assessed this in your study.

Line 299: Minor, but consider deleting “devastating” here or in the previous line 296

Line 300: This information is not clear, and Im not sure is relevant as it is included. Do you mean that there were reports that animals had died because of the application of ivermectin?

Discussion

Line 310: I suggest not using “confirmed” here. Malaria diagnosis can be based on clinical findings and can be also confused with other febrile illness (even some transmitted by mosquitoes, such as dengue) and I don’t think is accurate to say that your participants can confirm that malaria is a very common disease, as confirmation would need laboratory results. Suggest modifying the language.

Line 317: This is an important finding in this context, but in many cases, children are the “right hosts” for interventions, so I don’t think it is accurate to say that having adults instead of children is a basic element of research globally. Suggest modifying the language here.

Line 359: This further emphasizes the convenience of including additional information on the specific timing of your activities and the occurrence of additional information activities before the study implementation

Another limitation of your study is that it seems to have been conducted along the implementation of the SLIM trial, but you did not have the chance to assess community perceptions before, hence the applicability to the planning phase of other studies can be limited.

Line 379: Minor but consider changing “must”. New vector control tools for Anopheles are needed?

Line 379: the sentence starting with “Two of the…” reads strange. Please review and consider modifying for readability.

Overall, the discussion section can benefit from additional content on how your results can improve the implementation of research in Sumba, the description and conclusions seem very general and broad. Simple, regular, and respectful communication should be part of any community engagement effort; it would be very valuable to get more specific information on how you think your results can inform the implementation of additional malaria research around ivermectin.

Reviewer #2: General Comment:

Mary and colleagues’ study on community perceptions of ivermectin for malaria control on Sumba Island is both timely and well-articulated. I commend the authors for this thoughtful and engaging work. I have a few overall and specific comments, primarily aimed at enhancing readability and encouraging additional engagement with relevant literature, particularly from similar interventions in Southeast Asia.

Abstract

If possible, please avoid using too many acronyms in the abstract. Terms like ITL forced me to go back and forth while reading. You don’t want readers to feel interrupted.

Methods

You mention using mixed methods, but only describe the qualitative data analysis. Please include details about the quantitative or other methods used as well.

Also, consider briefly explaining what NVivo is for readers who may not be familiar with it.

Results

The phrase trust-building efforts feels vague. Can you provide specific examples to make this clearer?

Similarly, low social cohesion posed barriers would benefit from an illustrative example.

You introduce the term pre-trial here without having explained the category structure of pre-, during-, and post-trial stages earlier in the manuscript. For a broader audience unfamiliar with engagement-specific terminology, it might be clearer to say something like before the trial or as early as possible.

Conclusions

You raise some compelling points in your results — for instance, about children’s participation, social cohesion, and engagement with authorities. Why not make the conclusion section slightly more specific? As you know, terms like trust, engagement, and cultural considerations are often overused. You could strengthen your conclusions by pointing out how you specifically addressed these — not just through engagement in general, but through what aspects of engagement (e.g., dialogue, informal interactions, mediators)? You've discussed this effectively in the main text — you could reflect that clarity here too.

Specific Lines

• Line 119: Did you mean written informed consent?

• Line 139: You previously mentioned 75 participants, but here you say 24 transcripts. Can you clarify the discrepancy?

• Somewhere in the data collection section, could you add a table or box summarizing the types of data collected and the number of participants involved in each? I see you have table 1, but something simple, for readers to give an idea of types and breath of data collection, without having to navigate the acronyms.

• Line 303–307: The results related to low social cohesion is quite interesting, and quite unique to the research site you have collected data from. Do you have more elaboration or data to support this?

Discussion: Knowledge Deficit Model (Lines 328–337)

I appreciate your argument regarding the knowledge deficit model — in essence, challenging the idea that community members are "deficient" in understanding. But concepts like MDA are genuinely complex. In many of our studies across Southeast Asia, people have questioned the rationale: Why take medicine when you're apparently healthy?

This isn't ignorance — it’s a natural and valid response. Some of your co-authors have long studied how MDA is perceived, especially in the context of asymptomatic malaria. As you rightly emphasize, the answer isn’t to preach but to communicate with respect and clarity. Still, informed participation requires explanation — what is MDA, how does it work, and why is it necessary? Trust (but not blind placed trust) matters, yes, but so does making the concept understandable — that’s the essence of meaningful informed consent, isn’t it?

Additional Reflection: Social Cohesion

As a collaborator of multi-disciplinary research, we also found lack of social cohesion to be a major factor in MDA participation — particularly along the Thai–Myanmar border. Political divisions within communities, or tensions between ethnic groups, heavily influenced community responses. In contrast, settings like Laos and Cambodia, with stronger cohesion, saw greater participation. You might find that experience relevant and worth integrating into your discussion.

**Do you want your identity to be public for this peer review?** For information about this choice, including consent withdrawal, please see our Privacy Policy

Reviewer #1: No

Reviewer #2: No

---

## [Author Response · Author response to Decision Letter 1]

6 Sep 2025

Dear Dr Pyae Linn Aung and reviewers,

Thank you for the thoughtful review of our manuscript: Community perceptions and acceptance of ivermectin for malaria control on Sumba Island, Indonesia. We have revised the manuscript accordingly and have responded to the journal and reviewer suggestions below. We believe that these changes have substantially improved the paper.

Best wishes,

Mary Chambers on behalf of authors

Response to Journal Requirements:

1.Please ensure that your manuscript meets PLOS ONE's style requirements.

We have edited accordingly.

2. We note that you have indicated that there are restrictions to data sharing for this study.

Data is available by request of OUCRU Data Access Committee: CTU@oucru.org

The de-identified card-sorting data (Table 2) has been uploaded to Zenodo: https://doi.org/10.5281/zenodo.16992379

Please update our Data Availability statement to reflect this information.

3. We note that Figure 1 in your submission contain map images which may be copyrighted.

Figure 1 was created in QGIS by the authors using administrative boundary data from GADM version 4.1. We have added this detail to the figure title.

4. Please remove all personal information, ensure that the data shared are in accordance with participant consent, and re-upload a fully anonymized data set.

No personal information has been included in the manuscript. The local names are names of villages.

We have added captions for the supporting information, and in the text (line 219)

We have included new references relevant to this manuscript.

Response to reviewers' comments to the authors:

We are grateful to both reviewers for their careful review of this manuscript, and for their helpful suggestions. Our responses to each comment is given in italics below.

Reviewer #1: This paper by Chambers et al. describes the results of qualitative research conducted along an ivermectin treatment of livestock (ITL) in Sumba, Indonesia. Overall, additional information on certain aspects of the methods, and a more detailed conclusion section on how the results can be used to improve research in this context, would benefit the manuscript.

We have added more details to the methods and discussion.

Introduction

The introduction is very well written and very informative.

Line 88: minor but suggest grouping this information (the specific number of cases) with the information in line 60.

Have moved to Line 60

Study design

It would be helpful to include very clear information on the timing of this study related to the SLIM project. Although implied that you did not work before the implementation of SLIM, it is only in the discussion section where this information is specified.

See Line 126 -130 (revised MS). We have made this clearer.

Minor, but suggest adding “author” to all instances where the authors initials are used

Have made this amendment in some places. Have kept initials when it relates to data collection and analysis following some social science convention.

Results

Line 170: school children were all participants up to 18? 21? And then anyone over that age was considered adult? Seems that only having two categories here “school children” and “adults” would leave some participants out? I suggest adding information to clarify

Line 197 (revised MS). We have added that information

It would be interesting to know the age distribution of your participants. Were there on the older side? Age can be a key determinant for perception on ‘new’ methods to control mosquitoes, and it would be helpful to know if all your participants were younger vs older adults.

We didn’t collect that data, however most FGD and IDI participants were >30 years old as they were working as cadres or were cattle owners.

Table 1: Add the total N to the table. Make sure that the title of the table is complete, including dates, and place, so that stands alone.

We have edited the title

Line 198: Consider moving this to the discussion section

We have moved this sentence

Line 227: More information on informational/educational campaigns that took place before the implementation of SLIM would be helpful to give more context to this part of the study. Was there a broad community outreach effort before this study?

No, the study villages are geographically far apart and so broad community outreach was not possible Engagement with district and sub-district officials was conducted prior to the study starting. However community outreach in each village was conducted as we selected the next field site. Site selection could not be done months in advance as it depended on mosquito populations at the time of the study. Sensitization for the community was conducted after the FGDs and IDIs so as not to influence people’s perceptions about malaria and research.

We have added a paragraph explaining this (from line 126 of revised MS).

Line 233: this seems to be a very important finding, that animal owners trust government veterinarians and their involvement in projects like these could improve participation, but I don’t think this is highlighted in the discussion. Government involvement is not always a factor that increases participation in research studies, and this would be an important context-specific finding.

This is an important point, and we added a paragraph about this to the discussion.

Line 235: It is not clear to me what do you mean by “further into the trial”. Do you mean that in activities conducted at a later time, you got these findings?

We removed this phrase.

Line 263: This study was conducted after the COVID-19 pandemic. Was ivermectin use also widely discussed in media and social media in Indonesia? If yes, did you get any insight on participants perceptions associated with the use of ivermectin for covid and how that translate to MDA?

We have added the sentence “There was little Indonesian media discussion of ivermectin as a Covid-19 drug during the pandemic.”

Line 280: Just curious, I understand that you are translating directly from participants’ quotes, but was “drunk on drugs” the most accurate translation? Did they mean that drugs could be toxic for them? Or is being “drunk on drugs” a common saying in your population?

We have edited the translation to be clearer in the meaning.

Line 283: This section doesn’t seem to be something that you assessed through your study activities, but rather a description of some situations observed during the activities conducted to recruit participants for the SLIM study. When I read the section where you describe the topics, I thought you were going to describe results of specific questions asked to participants regarding what they perceive as barriers for implementing malaria research. I suggest modifying your results section to clarify that this is additional information but not part of your study questions/assessment or to modify language to clarify how you assessed this in your study.

Participant observation was a data collection method. We have made this clearer at the start of this section.

Line 299: Minor, but consider deleting “devastating” here or in the previous line 296.

We removed one use of this word.

Line 300: This information is not clear, and Im not sure is relevant as it is included. Do you mean that there were reports that animals had died because of the application of ivermectin?

We have clarified.

Discussion

Line 310: I suggest not using “confirmed” here. Malaria diagnosis can be based on clinical findings and can be also confused with other febrile illness (even some transmitted by mosquitoes, such as dengue) and I don’t think is accurate to say that your participants can confirm that malaria is a very common disease, as confirmation would need laboratory results. Suggest modifying the language.

We have changed the wording.

Line 317: This is an important finding in this context, but in many cases, children are the “right hosts” for interventions, so I don’t think it is accurate to say that having adults instead of children is a basic element of research globally. Suggest modifying the language here.

Have modified the language, but we are reporting the community view, not our recommendation.

Line 359: This further emphasizes the convenience of including additional information on the specific timing of your activities and the occurrence of additional information activities before the study implementation. Another limitation of your study is that it seems to have been conducted along the implementation of the SLIM trial, but you did not have the chance to assess community perceptions before, hence the applicability to the planning phase of other studies can be limited.

We did conduct pre-study data collection and a sensitization meeting in each village. We have added a sentence to make this process clearer. We appreciate the reviewer’s perspective that a stand-alone study would be more useful, however by conducting it alongside the entomology study, we were able to gather real-time data on community perceptions of the trial conduct. We see this as an advantage not a limitation.

Line 379: Minor but consider changing “must”. New vector control tools for Anopheles are needed?

We have changed the wording.

Line 379: the sentence starting with “Two of the…” reads strange. Please review and consider modifying for readability.

Revised.

Overall, the discussion section can benefit from additional content on how your results can improve the implementation of research in Sumba, the description and conclusions seem very general and broad. Simple, regular, and respectful communication should be part of any community engagement effort; it would be very valuable to get more specific information on how you think your results can inform the implementation of additional malaria research around ivermectin.

We have added more specific recommendations.

Reviewer #2: General Comment:

Mary and colleagues’ study on community perceptions of ivermectin for malaria control on Sumba Island is both timely and well-articulated. I commend the authors for this thoughtful and engaging work. I have a few overall and specific comments, primarily aimed at enhancing readability and encouraging additional engagement with relevant literature, particularly from similar interventions in Southeast Asia.

Abstract

If possible, please avoid using too many acronyms in the abstract. Terms like ITL forced me to go back and forth while reading. You don’t want readers to feel interrupted.

We have amended in the abstract.

Methods

You mention using mixed methods, but only describe the qualitative data analysis. Please include details about the quantitative or other methods used as well.

Also, consider briefly explaining what NVivo is for readers who may not be familiar with it.

We removed the reference to mixed methods and we added to explain NVIVO “a qualitative data analysis software package”

Results

The phrase trust-building efforts feels vague. Can you provide specific examples to make this clearer?

Yes, we agree this should be unpacked. We have added examples.

Similarly, low social cohesion posed barriers would benefit from an illustrative example.

This is now explained fully in the main text, with references to other studies in the region that found similar findings in the discussion (Line 418 revised MS).

You introduce the term pre-trial here without having explained the category structure of pre-, during-, and post-trial stages earlier in the manuscript. For a broader audience unfamiliar with engagement-specific terminology, it might be clearer to say something like before the trial or as early as possible.

We have edited the wording

Conclusions in abstract

You raise some compelling points in your results — for instance, about children’s participation, social cohesion, and engagement with authorities. Why not make the conclusion section slightly more specific? As you know, terms like trust, engagement, and cultural considerations are often overused. You could strengthen your conclusions by pointing out how you specifically addressed these — not just through engagement in general, but through what aspects of engagement (e.g., dialogue, informal interactions, mediators)? You've discussed this effectively in the main text — you could reflect that clarity here too.

We have added more details (Line 47, revised MS)

Specific Lines

• Line 119: Did you mean written informed consent? Yes, amended

• Line 139: You previously mentioned 75 participants, but here you say 24 transcripts. Can you clarify the discrepancy? The four FGDs had between 10-18 people but only generated 1 transcript each discussion.

• Somewhere in the data collection section, could you add a table or box summarizing the types of data collected and the number of participants involved in each? I see you have table 1, but something simple, for readers to give an idea of types and breath of data collection, without having to navigate the acronyms.

Agree. We have removed some of the acronyms in this section.

• Line 303–307: The results related to low social cohesion is quite interesting, and quite unique to the research site you have collected data from. Do you have more elaboration or data to support this?

This information was given by a team member who was local to that area. We do not have more details or data. We have added that source of the information (line 348 revised MS).

Discussion: Knowledge Deficit Model (Lines 328–337)

I appreciate your argument regarding the knowledge deficit model — in essence, challenging the idea that community members are "deficient" in understanding. But concepts like MDA are genuinely complex. In many of our studies across Southeast Asia, people have questioned the rationale: Why take medicine when you're apparently healthy?

This isn't ignorance — it’s a natural and valid response. Some of your co-authors have long studied how MDA is perceived, especially in the context of asymptomatic malaria. As you rightly emphasize, the answer isn’t to preach but to communicate with respect and clarity. Still, informed participation requires explanation — what is MDA, how does it work, and why is it necessary? Trust (but not blind placed trust) matters, yes, but so does making the concept understandable — that’s the essence of meaningful informed consent, isn’t it?

This is an important point, and we have added a few sentences and referred to good practise examples of engagement supporting MDA.

Additional Reflection: Social Cohesion

As a collaborator of multi-disciplinary research, we also found lack of social cohesion to be a major factor in MDA participation — particularly along the Thai–Myanmar border. Political divisions within communities, or tensions between ethnic groups, heavily influenced community responses. In contrast, settings like Laos and Cambodia, with stronger cohesion, saw greater participation. You might find that experience relevant and worth integrating into your discussion.

Thank you, we have drawn on insights from other studies in the region that had similar findings. We feel that including these in the discussion has made it stronger and thank the reviewers for their suggestions.

---

## [Decision Letter · Decision Letter 1]

21 Oct 2025

Dear Dr. Chambers,

Thank you for submitting your manuscript to PLOS ONE. After careful consideration, we feel that it has merit but does not fully meet PLOS ONE’s publication criteria as it currently stands. Therefore, we invite you to submit a revised version of the manuscript that addresses the points raised during the review process.

We look forward to receiving your revised manuscript.

Kind regards,

Pyae Linn Aung

Academic Editor

PLOS ONE

Journal Requirements:

Additional Editor Comments :

Thank you for revising the manuscript in response to the reviewers’ comments. As one of the reviewers is unavailable for the second round, I have read the revised version from an editorial perspective. Please see my comments below.

1. In the abstract, the Methods section should specify the target population and indicate the timeline of data collection (e.g., before or after the trial). The total sample size (n = 75) should be clearly categorized by data collection method (e.g., FGDs, IDIs, etc.).

2. In the Introduction, it would be useful to acknowledge the recent resurgence of malaria in Indonesia. Additionally, please clarify that different Anopheles species exhibit varied host-seeking behaviors, some are zoophilic, while others are anthropophilic, given that the intervention in this trial targeted only animals.

3. The Methodology section requires particular attention. The map should include a north arrow for clarity. The sampling strategy is unclear: please explain how participants were selected for each data collection method to ensure representativeness across study areas. Line 141 mentions gender balance, yet the majority of participants were male; moreover, Lines 148–149 indicate that women were uncomfortable participating in FGDs, which further limited female representation. This suggests that the findings primarily reflect male perspectives. Please describe the tools used for each data collection method, and consider attaching them as supplementary materials. Explain how responses were recorded, given that only one researcher conducted the data collection with assistance from a translator. Clarify the process of language translation and ensure the NVivo software is properly cited.

4. Line 202: Participants included government officials, health cadres, community leaders, and livestock owners. Given their differing social roles and backgrounds, please explain how you ensured that all voices were equally represented during discussions.

5. Line 204: Please clarify what is meant by “their experience.” For example, does this refer to years of livestock farming or another indicator?

6. Although multiple qualitative methods were employed, the manuscript contains very few participant quotes. Including more direct quotations would strengthen the findings and illustrate key themes. Given the diversity of participants’ socioeconomic and demographic backgrounds, it would also be useful to identify the speaker type (e.g., community livestock owner, government official) when quoting, as their perspectives likely differ.

7. Lines 290–310 discuss ivermectin MDA for humans. Please clarify how you ensured that participants understood the purpose of the drug when applied to humans, particularly since they observed its use in animals. This could have caused confusion about whether a drug given to animals is safe or appropriate for human use.

8. Several sentences should be reviewed for technical accuracy and precision. For example, Lines 24–25: not all Anopheles species transmit malaria. Lines 68–69: the claim that having many Anopheles species makes vector control difficult should be reconsidered or supported with evidence.

9. All references need to be revised to conform to the journal’s formatting requirements, both in the reference list and in-text citations. For instance, see Lines 371 and 433.

Reviewers' comments:

Reviewer's Responses to Questions

**Comments to the Author**

Reviewer #2: All comments have been addressed

2. Is the manuscript technically sound, and do the data support the conclusions?

Reviewer #2: Yes

3. Has the statistical analysis been performed appropriately and rigorously?

Reviewer #2: N/A

4. Have the authors made all data underlying the findings in their manuscript fully available?

Reviewer #2: Yes

5. Is the manuscript presented in an intelligible fashion and written in standard English?

Reviewer #2: Yes

Reviewer #2: (No Response)

**Do you want your identity to be public for this peer review?** For information about this choice, including consent withdrawal, please see our Privacy Policy

Reviewer #2: No

---

## [Author Response · Author response to Decision Letter 2]

12 Jan 2026

12 January 2026

Pyae Linn Aung

Academic Editor

PLOS ONE

PONE-D-25-10247: Community perceptions and acceptance of ivermectin for malaria control on Sumba Island, Indonesia

Dear Dr Pyae Linn Aung,

Thank you for the thoughtful review of our manuscript: Community perceptions and acceptance of ivermectin for malaria control on Sumba Island, Indonesia. We have revised the manuscript accordingly and have responded to the Editor’s suggestions below. We believe that these changes have substantially improved the paper.

Best wishes,

Mary Chambers on behalf of authors

Manuscript edits. Line numbers correspond to the Revised manuscript with track changes.

1. In the abstract, the Methods section should specify the target population and indicate the timeline of data collection (e.g., before or after the trial). The total sample size (n = 75) should be clearly categorized by data collection method (e.g., FGDs, IDIs, etc.).

Edited as suggested.

2. In the Introduction, it would be useful to acknowledge the recent resurgence of malaria in Indonesia. Additionally, please clarify that different Anopheles species exhibit varied host-seeking behaviors, some are zoophilic, while others are anthropophilic, given that the intervention in this trial targeted only animals.

We have acknowledged the resurgence in malaria in Indonesia (line 65), and note the differing feeding patterns of Anophelese species. However, please note that this manuscript is focusing on the social science aspect of the study and the entomology data is presented in our first paper: Kobylinski, K.C., et al., Impact of standard and long-lasting ivermectin formulations in cattle and buffalo on wild Anopheles survival on Sumba Island, Indonesia. Scientific Reports, 2024. 14(1): p. 29770.

3. The Methodology section requires particular attention. The map should include a north arrow for clarity. The sampling strategy is unclear: please explain how participants were selected for each data collection method to ensure representativeness across study areas. Line 141 mentions gender balance, yet the majority of participants were male; moreover, Lines 148–149 indicate that women were uncomfortable participating in FGDs, which further limited female representation. This suggests that the findings primarily reflect male perspectives. Please describe the tools used for each data collection method, and consider attaching them as supplementary materials. Explain how responses were recorded, given that only one researcher conducted the data collection with assistance from a translator. Clarify the process of language translation and ensure the NVivo software is properly cited.

Edits made as suggested and the gender of participants sampled by different methods is made clearer.

Tools have been made available https://doi.org/10.5281/zenodo.17769585

4. Line 202: Participants included government officials, health cadres, community leaders, and livestock owners. Given their differing social roles and backgrounds, please explain how you ensured that all voices were equally represented during discussions.

Edited as suggested. Line 193

5. Line 204: Please clarify what is meant by “their experience.” For example, does this refer to years of livestock farming or another indicator?

Edited as suggested. Line 264

6. Although multiple qualitative methods were employed, the manuscript contains very few participant quotes. Including more direct quotations would strengthen the findings and illustrate key themes. Given the diversity of participants’ socioeconomic and demographic backgrounds, it would also be useful to identify the speaker type (e.g., community livestock owner, government official) when quoting, as their perspectives likely differ.

Edited as suggested and more quotes have been included.

7. Lines 290–310 discuss ivermectin MDA for humans. Please clarify how you ensured that participants understood the purpose of the drug when applied to humans, particularly since they observed its use in animals. This could have caused confusion about whether a drug given to animals is safe or appropriate for human use.

Edited as suggested. Line 378-386.

8. Several sentences should be reviewed for technical accuracy and precision. For example, Lines 24–25: not all Anopheles species transmit malaria. Lines 68–69: the claim that having many Anopheles species makes vector control difficult should be reconsidered or supported with evidence.

Edited as suggested and made clearer.

9. All references need to be revised to conform to the journal’s formatting requirements, both in the reference list and in-text citations. For instance, see Lines 371 and 433.

Revised as suggested.

---

## [Editor Report · Decision Letter 2]

20 Jan 2026

Community perceptions and acceptance of ivermectin for malaria control on Sumba Island, Indonesia

PONE-D-25-10247R2

Dear Dr. Chambers,

We’re pleased to inform you that your manuscript has been judged scientifically suitable for publication and will be formally accepted for publication once it meets all outstanding technical requirements.

Kind regards,

Pyae Linn Aung

Academic Editor

PLOS One

Additional Editor Comments (optional):

Thank you to the authors for improving the manuscript in response to my comments. For me, it is now acceptable, although I still notice some typographical errors and inconsistencies in reference formatting. Please review and correct these during the proofing stage, especially since PLOS does not provide editorial editing support.
---

## [Editor Report · Acceptance letter]

PONE-D-25-10247R2

PLOS One

Dear Dr. Chambers,

I'm pleased to inform you that your manuscript has been deemed suitable for publication in PLOS One. Congratulations! Your manuscript is now being handed over to our production team.

Kind regards,

on behalf of

Dr. Pyae Linn Aung

Academic Editor

PLOS One